# DOT1L Methyltransferase Regulates Calcium Influx in Erythroid Progenitor Cells in Response to Erythropoietin

**DOI:** 10.3390/ijms23095137

**Published:** 2022-05-05

**Authors:** Yi Feng, Shaon Borosha, Anamika Ratri, Eun Bee Lee, Huizhen Wang, Timothy A. Fields, William H. Kinsey, Jay L. Vivian, M. A. Karim Rumi, Patrick E. Fields

**Affiliations:** 1Department of Pathology and Laboratory Medicine, University of Kansas Medical Center, Kansas City, KS 66160, USA; yfeng1@email.unc.edu (Y.F.); sborosha2@gmail.com (S.B.); aratri@kumc.edu (A.R.); elee10@kumc.edu (E.B.L.); tfields@kumc.edu (T.A.F.); jvivian@kumc.edu (J.L.V.); mrumi@kumc.edu (M.A.K.R.); 2Department of Anatomy and Cell Biology, University of Kansas Medical Center, Kansas City, KS 66160, USA; hwang@kumc.edu (H.W.); wkinsey@kumc.edu (W.H.K.)

**Keywords:** DOT1L, erythroid progenitors, erythropoietin, TRPC6, calcium influx

## Abstract

Erythropoietin (EPO) signaling plays a vital role in erythropoiesis by regulating proliferation and lineage-specific differentiation of murine hematopoietic progenitor cells (HPCs). An important downstream response of EPO signaling is calcium (Ca^2+^) influx, which is regulated by transient receptor potential channel (TRPC) proteins, particularly TRPC2 and TRPC6. While EPO induces Ca^2+^ influx through TRPC2, TRPC6 inhibits the function of TRPC2. Thus, interactions between TRPC2 and TRPC6 regulate the rate of Ca^2+^ influx in EPO-induced erythropoiesis. In this study, we observed that the expression of TRPC6 in KIT-positive erythroid progenitor cells was regulated by DOT1L. DOT1L is a methyltransferase that plays an important role in many biological processes during embryonic development including early erythropoiesis. We previously reported that *Dot1l* knockout (*Dot1l^KO^*) HPCs in the yolk sac failed to develop properly, which resulted in lethal anemia. In this study, we detected a marked downregulation of *Trpc6* gene expression in *Dot1l**^KO^* progenitor cells in the yolk sac compared to the wild type (WT). The promoter and the proximal regions of the *Trpc6* gene locus exhibited an enrichment of H3K79 methylation, which is mediated solely by DOT1L. However, the expression of *Trpc2*, the positive regulator of Ca^2+^ influx, remained unchanged, resulting in an increased TRPC2/TRPC6 ratio. As the loss of DOT1L decreased TRPC6, which inhibited Ca^2+^ influx by TRPC2, *Dot1l^KO^* HPCs in the yolk sac exhibited accelerated and sustained elevated levels of Ca^2+^ influx. Such heightened Ca^2+^ levels might have detrimental effects on the growth and proliferation of HPCs in response to EPO.

## 1. Introduction

The regulation of erythropoiesis is a highly orchestrated process involving epigenetic and transcriptional regulation in response to a number of cytokines and growth factors [1]. Amongst all external factors, erythropoietin (EPO) is considered the primary regulator of erythropoiesis [2]. Loss of function of either EPO or the EPO receptor (EPOR) indicates that this interaction is crucial for definitive erythropoiesis in the embryo, and developing mice become severely anemic and die by embryonic day 13.5 (E13.5) [3,4]. When EPO binds to EPO receptors, several signaling pathways are initiated, particularly the phospholipase Cγ (PLCγ) signaling pathway [1]. Activation of PLCγ generates the second messenger, inositol-1,4,5-trisphosphate (IP3). A conformational change in the IP3 receptor upon IP3 binding facilitates its association with transient receptor potential channel (TRPC) proteins, which are voltage-independent calcium (Ca^2+^) channels [1,5,6,7]. TRPC proteins mediate Ca^2+^ influx into erythroid progenitors [7,8]. It has been demonstrated that regulation of intracellular Ca^2+^ by EPO plays a critical role in the survival, proliferation, and differentiation of erythroid progenitors [9,10]. Among the six members of the mouse TRPC family, TRPC2 and TRPC6 messenger RNAs (mRNAs) and proteins are expressed in erythropoietic cell lines [7,8] as well as primary murine erythroid cells [7,8,11]. In addition, these proteins can interact to form heteromultimeric channels [11]. Multimeric channel formation has been reported for many other TRPC family members [12,13,14,15,16] and, in at least one other case, formation of this heteromultimeric channel resulted in the cation channel possessing properties that were substantially different from those of the homomeric channel [12]. In erythroid cells, EPO stimulation induces Ca^2+^ influx through TRPC2 homomeric channels, but when the channels are heteromultimeric and incorporate TRPC6, EPO-induced calcium influx through TRPC2 is attenuated [11]. Thus, it is speculated that the interaction of TRPC2 and TRPC6 plays an important role in erythroid cells to regulate Ca^2+^ influx in response to EPO stimulation.

In eukaryotic cells, DNA is packaged within the nucleus along with histones and other nuclear proteins to form the nucleosome, the fundamental repeating unit of chromatin. The histones can be post-translationally modified in a variety of ways including acetylation, phosphorylation, ubiquitination, and methylation. These modifications influence chromatin structure, facilitate interactions between nucleosomes, and can potentially regulate transcription [17]. Methylation on lysine (K), which is one of the covalent histone modifications, exists in mono, di, and tri states. This modification is found predominantly on lysine residues at the N-terminal tails of histones H3 and H4 and is catalyzed by one of the family members of histone lysine methyltransferases (HLMTs) [18]. Most HLMT family members contain a conserved SET (suppressor of variegation, enhancer of zeste, and trithorax) domain, which is required for enzymatic activity. In contrast, disruptor of telomeric silencing 1-like (DOT1L), a histone methyltransferase that targets lysine 79 of histone H3 (H3K79), is quite different from most other HLMT family members [19]. DOT1 family members do not have a SET domain, and its substrate, K79, is located within the globular domain of histone H3 [20,21]. DOT1L is the only known methyltransferase in eukaryotic cells responsible for mono-, di-, and tri-methylation of H3K79 [21], and these histone modifications are strongly associated with actively transcribed chromatin regions [22]. The enzyme was first described in yeast to play an important role in telomere silencing [23]. It is also required for the DNA damage response and is associated with gene transcription activity [22,24]. After the DOT1L-deficient mouse line was established, it was shown that DOT1L is required for mouse embryonic development [19], and has since been shown to play a crucial role in many embryonic developmental processes [21,25,26]. We previously reported that DOT1L deficiency in knockout (*Dot1l^KO^*) mice results in an embryonic erythropoietic defect and embryonic lethality during mid-gestation in these mice [25]. Consistent with our results about DOT1L function in hematopoiesis, two other groups using a conditional *Dot1l^KO^* model showed that DOT1L is also required for murine postnatal hematopoiesis [27,28].

*Dot1l^KO^* erythroid progenitors failed to develop normally, showing cell cycle arrest and increased apoptosis [25]. However, the molecular mechanisms underlying DOT1L regulation of early erythropoiesis remain unclear. In this study, we sorted KIT-positive cells from yolk sacs of E10.5 mice in order to enrich for yolk-sac-derived hematopoietic progenitor cells. We found that *Trpc6* is a direct target of DOT1L in these cells. We detected an enrichment of H3K79 methylation within the *Trpc6* gene locus. Moreover, the loss of *Trpc6* expression in *Dot1l^KO^* HPCs showed accelerated and sustained high levels of Ca^2+^ influx and correlated with erythroid progenitor cell cycle arrest and death.

## 2. Results

### 2.1. Dot1l^KO^ Erythroblasts Displayed Decreased Proliferation, Cell Cycle Arrest, and Increased Apoptosis

Embryos and yolk sacs were collected on embryonic day 10.5 (E10.5) (Figure 1A). E10.5 *Dot1l^KO^* embryos and yolk sacs were markedly smaller than the wild-type embryos and yolk sacs (Figure 1B,C). While the wild-type yolk sacs and embryos possessed prominent blood vessels and red blood cells, *Dot1l^KO^* yolk sacs and embryos were pale (Figure 1B,C). Equal numbers of cells isolated from the E10.5 wild-type and *Dot1l^KO^* yolk sacs were cultured in erythroblast expansion medium containing erythropoietin that differentiated the definitive HPCs into extensively self-renewing erythroblasts (ESREs) [29]. *Dot1l^KO^* erythroblasts showed severely blunted proliferation, which resulted in a reduced number of growing cells compared to the wild type (Figure 1D,E).

On day 4 of the ESRE cultures, cells were harvested for cell cycle analyses and analysis of apoptosis (Figure 2A). We detected that a large number of *Dot1l^KO^* erythroblasts underwent a G_0_/G_1_ cell cycle arrest, which resulted in significantly reduced numbers of cells in the S and G2/M phases (Figure 2B–D). Moreover, there was a significant increase in the percentage of *Dot1l^KO^* erythroid progenitors that expressed annexin V, indicative of cells undergoing apoptosis. (Figure 2E).

For the in vitro definitive erythropoietic assays, cells isolated from E10.5 yolk sacs were plated on methylcellulose media in the presence of appropriate cytokines for colony formation [30]. Cells from *Dot1l^KO^* yolk sacs formed significantly smaller erythroid (BFU-E) colonies than those of the wild type (Figure 3A,B). In contrast, the wild-type and *Dot1l^KO^* myeloid (CFU-GM) colonies were similar in size (Figure 3C,D). However, both erythroid and myeloid colonies grown from the *Dot1l^KO^* yolk sac cells appeared less dense than the wild-type colonies (Figure 3B,D).

### 2.2. TRPC6 Expression Was Downregulated in Dot1l^KO^ Hematopoietic Progenitor Cells

We analyzed the gene expression levels of *Trpc* family members in KIT-positive HPCs isolated from E10.5 wild-type and *Dot1l^KO^* yolk sacs. We observed that among the three *Trpc* members expressed in these cells (i.e., *Trpc1*, *Trpc2*, and *Trpc6*), only *Trpc6* levels were significantly reduced in *Dot1l^KO^* progenitor cells. Other TRPC members remained relatively unchanged (Figure 4A). We also detected a marked reduction in TRPC6 protein expression in *Dot1l^KO^* yolk sacs cells (Figure 4B,C). These data indicate that DOT1L deficiency leads to reduced expression of TRPC6 in mouse HPCs. Since TRPC2 levels are not affected, the TRPC2/TRPC6 ratio in these progenitor cells increased from 0.95 to 4.40. 

### 2.3. H3K79 Methylation Was Closely Associated with the Trpc6 Expression Level

We observed that in *Dot1l^KO^* progenitor cells, *Trpc6* mRNA levels were significantly reduced compared to the wild type (Figure 4A). We investigated whether the *Trpc6* expression level correlates with H3K79 di- and tri-methylation status. We chose mouse bone marrow cells as high *Trpc6*-expressing and liver cells as low *Trpc6*-expressing cells (Figure 5A). RT-PCR and Western blot analyses confirmed the expression levels of *Trpc6* in bone marrow and liver cells (Figure 5A,B). ChIP-qPCR assays were performed for H3K79me2 and H3K79me3 by using PCR primers designed to cover the whole *Trpc6* locus including the promoter region, transcription start site, and middle and end of the gene locus (Figure 5C). Our results demonstrated a significant enrichment of H3K79 di- and tri-methylation at the promoter region and transcription start site of *Trpc6* in bone marrow cells (Figure 5D,E). In the middle and end regions of the *Trpc6* gene locus, enrichment of H3K79 methylation in bone marrow cells was not different from that of the liver cells (Figure 5D,E).

### 2.4. Abnormal Calcium Influx in Dot1l^KO^ KIT-Positive HPCs in Response to EPO

We tested whether Ca^2+^ influx is affected in *Dot1l^KO^* progenitor cells upon EPO treatment. We used the E10.5 yolk sac as the source of progenitor cells and analyzed the KIT-positive cells for Ca^2+^ influx during a period of 20 min using a fluorescence microscopy-coupled digital imaging system (Figure 6). We observed that when the wild-type yolk sac cells were exposed to EPO, the intracellular Ca^2+^ concentration began to increase gradually after 5 min and reached a plateau approximately after 15 min. In sharp contrast, the Ca^2+^ levels of *Dot1l^KO^* yolk sac cells increased immediately after EPO stimulation and continued to increase throughout the observation period. After 20 min, the Ca^2+^ signal reached a level approximately two-fold that of the wild-type yolk sac cells (Figure 6).

## 3. Discussion

The premise of this study was that an abnormal EPO response occurs in *Dot1l^KO^* HPCs due to the decreased level of TRPC6 expression. Our data suggest that H3K79 methylation is disrupted in the absence of DOT1L, which is the underlying cause of decreased *Trpc6* expression in *Dot1l^KO^* HPCs. We recently generated a mouse model that specifically lacks the methyltransferase activity of DOT1L protein [30]. RNA-seq analysis of the methyl mutant yolk-sac-derived erythroblasts showed a nine-fold downregulation of *Trpc6* expression (FDR *p* < 0.00) [31] (PRJNA666736), consistent with the data presented in Figure 4. These data, coupled with our findings of enriched H3K79 di- and tri-methylation in the promoter and transcriptional start sites of the *Trpc6* locus (Figure 5), indicate that DOT1L-mediated H3K79 methylation is essential for *Trpc6* expression in HPCs. 

The decreased expression of TRPC6 increases the TRPC2/TRPC6 ratio. An increased TRPC2/TRPC6 ratio results in a prolonged and heightened level of Ca^2+^ influx in HPCs, in accordance with previously published findings [11]. We propose that the accentuated Ca^2+^ influx in response to aberrant EPO signaling plays an important role in determining the *Dot1l^KO^* erythropoietic phenotype [25,30]. However, our findings do not exclude the possibility that other DOT1L-regulated cellular mechanisms may contribute to the phenotypic abnormalities observed in *Dot1l^KO^* HPCs.

We observed that the first wave of primitive as well as the second wave of definitive HPCs were present in the *Dot1l^KO^* yolk sacs [25,30]. Although the development of progenitors was not impaired, loss of DOT1L severely affected their proliferation and lineage-specific differentiation [25,30]. Despite an increased responsiveness to EPO, *Dot1l^KO^* definitive HPCs failed to proliferate. It needs to be noted that the development of erythroid progenitors does not require EPO signaling [3,4]; however, for the proliferation and differentiation of definitive erythroid progenitors, EPO signaling is essential [32,33]. An aberrant EPO response, due to the decreased TRPC6 expression, resulted in a prolonged and heightened influx of intracellular Ca^2+^ in KIT-positive *Dot1l^KO^* definitive HPCs. An increased Ca^2+^ influx can result in an excessive protein kinase activation, leading to increased levels of reactive oxygen species (ROS) [34,35] and cytotoxicity [36,37], which did not favor normal cell cycle progression as shown in Figure 2. Lack of DOT1L also gave rise to a condition where the HPCs failed to generate the required number of erythroid cells in vivo, to support the survival of the *Dot1l^KO^* embryos [25,30].

We also observed that ex vivo culture of HPCs from E10.5 wild-type yolk sacs in the presence of EPO and cytokines resulted in a huge proliferation of definitive erythroblasts [25,31]. In contrast, the HPCs from E10.5 *Dot1l^KO^* yolk sacs underwent cell cycle arrest and apoptosis. The cultured HPCs failed to expand (Figure 1 and Figure 2) and did not form erythroid colonies (Figure 3) under the same culture conditions in the presence of EPO and cytokines. While EPO promoted proliferation of wild-type definitive erythroblasts, an opposite response was observed in *Dot1l^KO^* erythroblast cells. Our findings also suggest that stimulation with EPO leads to the cell cycle arrest and death of *Dot1l^KO^* erythroid progenitor cells instead of sustained self-renewal due to an aberrant EPO response. However, it was not possible to determine if removal of EPO from E10.5 *Dot1l^KO^* HPC culture would recover their cell cycle arrest and proliferation, because EPO signaling is essential for the proliferation of definitive erythroblasts. Further experiments with an inducible *Dot1l^KO^* model would clarify the issue. 

Downregulation of TRPC6 expression due to the lack of DOT1L methyltransferase is responsible for the aberrant EPO response resulting in accelerated and heightened Ca^2+^ influx in KIT-positive HPCs as suggested in previous studies [11]. Both mRNA and protein levels of TRPC6 were reduced in KIT-positive *Dot1l^KO^* HPCs. We observed that expression of *Trpc6* correlated with the level of *Dot1l* expression and, thus, H3K79 methylation. Significantly, the level of H3K79 enrichment was substantially higher in cells that expressed higher amounts of TRPC6 (i.e., bone marrow cells) compared to those that expressed lower amounts (i.e., whole liver) (Figure 5). We also detected an enrichment of di- and tri-methyl H3K79 in the promoter and proximal regions of the *Trpc6* gene locus near the transcription start site. DOT1L is the only known methyltransferase that meditates H3K79 methylation in mammalian cells. Based on these findings, we can conclude that *Trpc6* is a direct target of DOT1L in erythroid progenitor cells. 

TRPC proteins play an important role in regulating the rate and state of Ca^2+^ influx [11,38]. Among the TRPC family members, only the expression of TRPC6 was downregulated in *Dot1l^KO^* erythroid progenitor cells, while others, including TRPC2 levels, remained intact, resulting in a four-fold increase in the TRPC2/TRPC6 ratio. Previous studies have shown that interaction between TRPC2 and TRPC6 is essential for modulating EPO-induced Ca^2+^ signaling [11]. Consequently, we observed an accelerated and increased level of sustained Ca^2+^ entry in *Dot1l^KO^* erythroid progenitor cells (Figure 6). Previous studies have also shown that EPO signaling results in Ca^2+^ influx, which plays an important role in cell survival, proliferation, and regulation of differentiation [5,9,39] (Figure 7A). However, dysregulation of intracellular Ca^2+^ levels, due to the disruption of TRPC6 expression, may result in toxic effects (Figure 7B). EPO-induced Ca^2+^ influx induces protein kinase activation that mediates hematopoiesis. However, excessively high levels of Ca^2+^ influx can result in sustained levels of PI3K, ERK, and AKT activation, which may lead to cell cycle arrest or apoptosis (Figure 7B). Thus, a reduced level of TRPC6 expression due to the loss of DOT1L in erythroid progenitor cells may result in lethal anemia despite normal EPO signaling.

## 4. Materials and Methods

### 4.1. Mouse Lines and Isolation of Cells from Yolk Sac

The *Dot1l^KO^* mouse line was generated in our previous study [25] and maintained by continuous backcrossing into C57BL/6 strains. Heterozygous *Dot1l^KO^* male and female mice were set-up for timed mating. Pregnant females were euthanized at E10.5, and the conceptuses were collected and dissected under stereomicroscopic examination [30]. For all the assays in this study, HPCs were derived from yolk sacs on E10.5. Single-cell suspensions were obtained as described previously [25]. Briefly, yolk sacs were incubated in 0.1% collagenase at 37 °C for 30 min. Then, the yolk sacs were aspirated through 25 G needles and filtered through a 70 µM strainer. Genotyping was performed by PCR using DNA extracted from corresponding embryo tissues [25]. All animal experiments were performed in accordance with the protocols approved by the University of Kansas Medical Center (KUMC) Animal Care and Use Committee.

### 4.2. Assessment of Cell Proliferation, Cell Cycle Analyses, and Apoptosis Assays

Single-cell suspensions of E10.5 wild-type and *Dot1l^KO^* yolk sacs were cultured in vitro in ESRE media (StemPro34 media containing nutrient supplement; ThermoFisher Scientific, Waltham, MA, USA), 2 U/mL human recombinant EPO (PeproTech, Inc., East Windsor, NJ, USA), 100 ng/mL SCF (PeproTech, Inc.), 10 µM dexamethasone (MilliporeSigma, Saint Louis, MO, USA), 40 ng/mL IGF1 (PeproTech, Inc., East Windsor, NJ, USA) and penicillin–streptomycin (ThermoFisher Scientific, Waltham, MA, USA) [29] as described in our previous publication [31]. After 4 days, cultured cells were imaged for estimation of cell density. Prior to cell cycle studies, E10.5 HPCs were cultured in M3334 medium for 4 days to generate erythroid colonies. The cells were separated by gentle pipetting, fixed by adding cold 70% ethanol slowly to the cell suspensions and treated with RNase. Cells were then stained with propidium iodide and analyzed by flow cytometry core for cell cycle progression [25,26]. Separately, fixed erythroid cells were labeled with annexin V and then analyzed by flow cytometry for signs of apoptosis as described previously [25,26]. Flow cytometry was performed using a FACSCalibur (BD Biosciences, San Jose, CA, USA), and analyses of cytometric data were carried out using CellQuest Pro software (BD Biosciences, San Jose, CA, USA) [25,26] at the KUMC flow cytometry core.

### 4.3. Analysis of Definitive Erythropoiesis from Yolk Sac Cells

Definitive erythropoiesis, erythroid (BFU-E), and myeloid (CFU-GM) colony formation assays were performed as described previously [30]. Approximately equal numbers of dissociated cells from wild-type *Dot1l^KO^* yolk sacs were plated in 35 mm culture dishes in M3434 methylcellulose medium (StemCell Technologies, Seattle, WA, USA) containing cytokines SCF (100 ng/mL), EPO (2 U/mL), IL-3 (5 ng/mL), and IL-6 (5 ng/mL) (PeproTech, Cranbury, NJ, USA), which promoted definitive erythroid and myeloid colony formation [30]. The cells were cultured at 37 °C for 10 days and scored according to the manufacturer’s recommendations.

### 4.4. Isolation of KIT-Positive Yolk Sac Cells

Single-cell suspensions were prepared from E10.5 yolk sacs as described above, and KIT-positive cells were separated as reported previously [25,26]. Briefly, HPCs were incubated with an anti-KIT antibody conjugated to phycoerythrin cyanine (ThermoFisher Scientific, Waltham, MA, USA) at 4 °C for 30 min. Then, the KIT-positive cells were isolated by cell sorting using a BD FACS Aria cell sorter (BD Biosciences, San Jose, CA, USA) at the KUMC flow cytometry core.

### 4.5. RNA Extraction, cDNA Preparation, and RT-qPCR

Total RNA was extracted using TRIzol reagent (ThermoFisher Scientific, Waltham, MA, USA) following the manufacturer’s instructions. cDNAs were prepared from 1 µg of total RNA collected from each sample using the SuperScript VILO cDNA Synthesis Kit (ThermoFisher Scientific, Waltham, MA, USA). Quantitative real-time PCR (qPCR) was performed using Power SYBR Green Master Mix (ThermoFisher Scientific, Waltham, MA, USA) and ran on a 7500 real-time PCR system (ThermoFisher Scientific, Waltham, MA, USA). qPCR results were normalized to Rn18S expression and calculated by the comparative ΔΔCT method [40,41,42]. A list of the qPCR primer sequences used in this study is shown in Table 1 (http://pga.mgh.harvard.edu/primerbank/, accessed on 1 May 2011).

### 4.6. Western Blot Analyses

Western blot analyses of TRPC6 were performed following standard protocols. Cells from the whole yolk sac, bone marrow, or liver were lysed in Nonidet-P40 (NP-40) buffer (150 mM sodium chloride, 1.0% NP-40, 50 mM Tris, and pH 8.0) on ice for 30 min. Cell extracts were centrifuged at 12,000× *g* at 4 °C for 20 min. Supernatants were collected and protein concentrations were measured using the Bradford assay (BioRad, Hercules, CA, USA). Approximately 30 µg of each protein sample was mixed with NuPAGE SDS 4× sample buffer (ThermoFisher Scientific, Waltham, MA, USA) and were electrophoresed on a 10% precast gel in SDS running buffer (250 mM Tris base, 190 mM glycine, 0.1% SDS, and pH 8.3) (BioRad, Hercules, CA, USA). Proteins were transferred to PVDF membranes, and the membranes were blocked in 5% bovine serum albumin (BSA) overnight at 4 °C. Blocked membranes were incubated with 1:300 TRPC6 primary antibody (Abcam, Inc., Cambridge, UK) for 4 h at 4 °C with shaking. Membranes were washed 5 times with TBST buffer at 4 °C with shaking and incubated with 1:5000 secondary antibody (Goat anti-Mouse IRDye CW800) for 45 min at room temperature. Then, the membranes were washed another 5 times with TBST, and images were captured with an Odyssey CLx infrared image system (LI-COR Biosciences; Lincoln, NE, USA). The same membrane was stripped and incubated with an antibody against ACTB (MilliporeSigma, Saint Louis, MO, USA) as a loading control of the protein samples.

### 4.7. Extraction of Bone Marrow and Liver Cells

Bone marrow cells that expressed high levels of *Trpc6* [16] were isolated from adult male mouse femurs as described previously [16]. Briefly, skin and muscles from the pelvic and femoral bones were removed. The bones were cleaned, and the ends were cut off at each end. Bone marrows were expelled from the ends of the bone passing cold sterile RNAse-free PBS with a 5 mL syringe fitted with a 25 G needle. Then, the cell aggregates were dispersed, and marrow cells were separated by repeated aspiration with a 5 mL syringe attached with a 27 G needle, passing through a 70 µM strainer. For liver cell isolation, mouse liver tissues were dissected and minced into small pieces in the presence of cold sterile RNAse-free PBS. The tissue pieces were ground between glass slides and passed through 70 µM strainer. Strained bone marrow or liver cells were washed with cold, sterile RNAse-free PBS used for RNA or protein extraction or fixed in formaldehyde for ChIP assays.

### 4.8. ChIP Assay for H3K79 Di- and Tri-Methylation in Trpc6 Locus

Chromatin immunoprecipitation (ChIP) was carried out following standard protocols [43,44]. Briefly, for each assay, 10^7^ cells were resuspended in 10 mL IMDM medium containing 5% FBS and were fixed in 1% formaldehyde for 10 min. Glycine was added to a final concentration of 125 mM and incubated at room temperature with rotation for 10 min to neutralize formaldehyde. The cells were washed twice with cold PBS, lysed in 500 µL SDS lysis buffer with protease inhibitor cocktail (MilliporeSigma, Saint Louis, MO, USA), and incubated on ice for 10 min. The nuclear pellets were washed with a cold lysis buffer and sonicated using a VWR Branson Sonifier 250 probe sonicator (VWR International, Radnor, PA, USA) 4–6 times, 15 s each time, to reduce the chromatin length to between 200 and 1000 bp. Sonicated chromatin solutions were centrifuged at 10,000× *g* at 4 °C for 10 min. Clear supernatants were collected into 1.5 mL tubes, and ~500 µL chromatin solution was mixed with 75 µL Protein A Agarose/Salmon Sperm DNA and incubated at 4 °C with rotation overnight. The solution was centrifuged at 3000× *g* for 5 min. The supernatant was transferred to a new tube and diluted 10-fold in ChIP Dilution Buffer. Two micrograms of H3K79me2 or H3K79me3 antibody (Abcam, Inc., Cambridge, UK) was added to 1 mL of the diluted chromatin solution each, and another 1 mL was used as “input”. Two micrograms of normal rabbit IgG (BioSource International, Inc., Camarillo, CA, USA) was used as negative control.

Chromatin solutions were incubated at 4 °C with rotation for 4 h in the presence of specific antibodies. Then, 65 µL Protein A Agarose/Salmon Sperm DNA was added to each tube and incubated at 4 °C with rotation for another 2 h. The solutions were centrifuged at 3000× *g* for 5 min at 4 °C, and the supernatants were discarded. The Protein A Agarose beads were washed for 5 min on a rotating platform sequentially with 1 mL of each of the low-salt wash buffer, high-salt wash buffer, LiCl wash buffer, and 1XTE. Then, the chromatins were eluted with 250 µL elution buffers twice, and the elutes were combined. Twenty microliters of 5 M NaCl and 2 µL RNase were added to each eluted chromatin sample and incubated at 37 °C for 30 min to remove RNAs and then again at 65 °C for 4 h to decrosslink the ChIPed chromatin materials. The chromatin solutions were mixed with 10 µL of 0.5 M EDTA, 20 µL 1 M Tris-HCl, PH 6.5, and 2 µL of 10 mg/mL Proteinase K and incubated at 45 °C for 1 h. Finally, the chromatin DNA fragments were recovered by phenol/chloroform extraction and ethanol precipitation. Real-time PCR was performed to quantify precipitated DNA with the use of the 7500 real-time PCR system. The ChIP-qPCR primers used for analyzing the *Trpc6* locus are listed in Table 2.

### 4.9. Measuring Calcium Influx of Yolk Sac Cells in Response to EPO

A fluorescence microscopy-coupled digital imaging system was used to measure the Ca^2+^ concentration change of HPCs when treated with EPO. Fura2 was used as the detection fluorophore. After the whole yolk sacs were dissociated into single cells, the cells were washed 2–3 times with PBS to completely remove fetal bovine serum (FBS). HPCs were resuspended in 40 µL RPMI 1640 (MilliporeSigma, Saint Louis, MO, USA) without FBS. Cells were loaded into the center of a culture dish coated with poly-lysine and fixed for 10 min at room temperature. The medium was then removed, and the cells were stained with 1 µM Fura2 and Pluronic F127 (ThermoFisher Scientific, Waltham, MA, USA) (1:1 in volume) at 37 °C for 30 min. The staining solution was removed, and the cells were covered with 150 µL RPMI containing 10% FBS. Mouse EPO (R&D Systems, Minneapolis, MN, USA) was added to a final concentration of 10 units/mL. Then, the sample was analyzed immediately by confocal fluorescence microscopy on a Nikon TE2000U microscope (Nikon, Tokyo, Japan). Fura2-loaded cells were visualized by digital video imaging, and the fluorescence was quantitated using the intensity ratio of the emission (510 nm), which was measured following excitation at 340 nm divided by the emission following excitation at 380 nm. Each sample was measured over 20 min. After, the data were calculated and analyzed using Metamorph 6.1 (Universal Imaging Corp., Downingtown, PA, USA).

### 4.10. Statistical Analyses

Each experimental group consisted of a minimum of 6 samples. The experimental results are presented as the mean ± standard error (SE). The results were analyzed for one-way ANOVA, and the significance of the mean differences was determined by Duncan’s post hoc test with *p* < 0.05. All the statistical calculations were performed using SPSS 22 (IBM, Armonk, NY, USA).

## Figures and Tables

**Figure 1 ijms-23-05137-f001:**
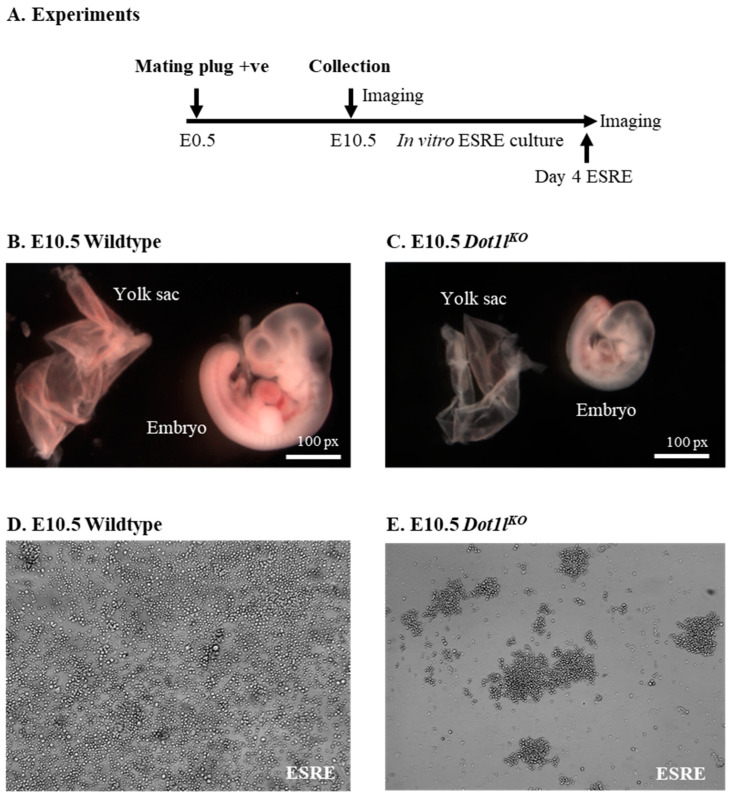
*Dot1l^KO^* progenitors failed to proliferate in response to EPO. (**A**) Schematic diagram showing the timeline of the embryo and yolk sac collection and in vitro culture of yolk-sac-derived erythroblasts. Representative images of E10.5 wild-type and *Dot1l^KO^* yolk sacs and embryos (**B**,**C**). Both the *Dot1l^KO^* yolk sacs as well as the embryos were smaller than those of the wild type (**B**,**C**). Cells were isolated from E10.5 wild-type and *Dot1l^KO^* yolk sacs and equal numbers were cultured in extensively self-renewing erythroblast (ESRE) culture medium as described. The cells were then differentiated from the definitive erythroid progenitors into erythroblasts (**D**,**E**). Compared to the wild-type erythroblasts (**D**), *Dot1l^KO^* erythroblasts had severely reduced numbers (**E**).

**Figure 2 ijms-23-05137-f002:**
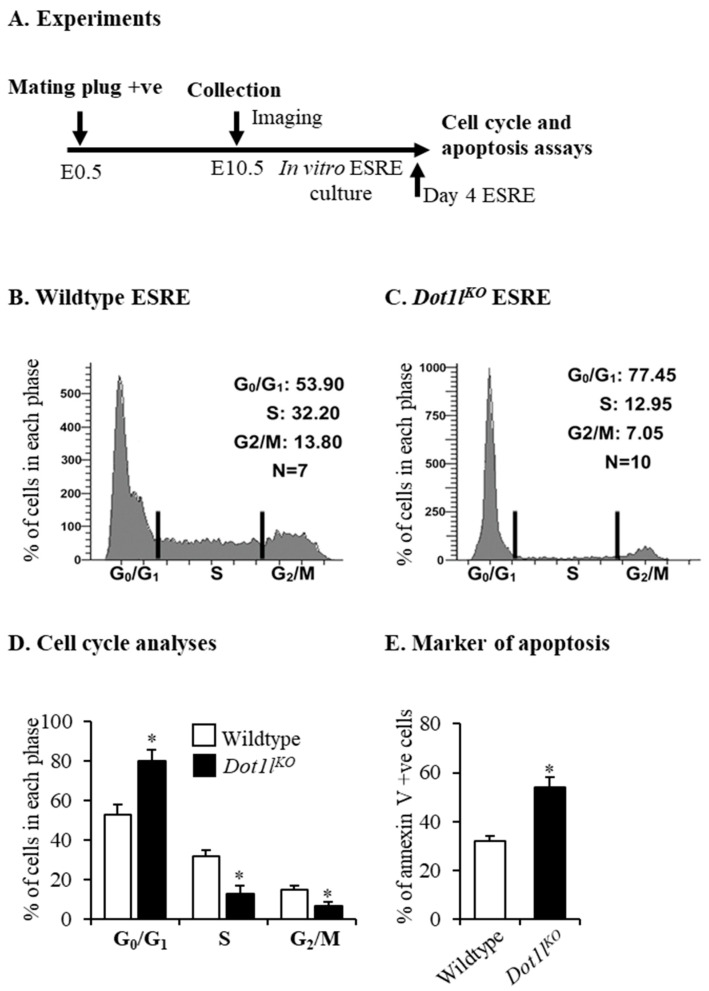
*Dot1l^KO^* progenitors underwent cell cycle arrest during in vitro erythroblast culture. (**A**) Schematic diagram showing the studies for cell cycle progression and apoptosis analysis. On day 4 of culture, cells were collected, fixed, and either stained with propidium iodide for cell cycle analyses or labeled with annexin V for signs of apoptosis. A large number of erythroid progenitors from *Dot1l^KO^* yolk sacs displayed G_0_/G_1_ arrest (**B**–**D**). In addition, an increased percentage of *Dot1l^KO^* erythroid progenitors were found to be annexin V positive (**E**). Data are expressed as the mean ± SE; *n* > 6; * *p* < 0.05.

**Figure 3 ijms-23-05137-f003:**
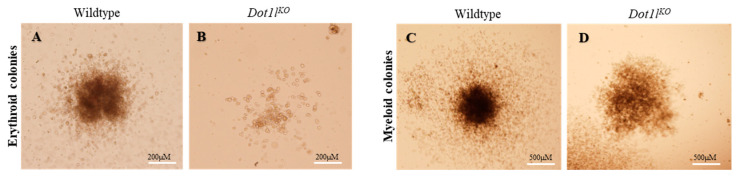
Equal numbers of wild-type and *Dot1l^KO^* E10.5 yolk sac cells were cultured in methylcellulose medium containing cytokines that promoted definitive erythroid and myeloid lineage differentiation. While *Dot1l^KO^* progenitors formed significantly smaller sized erythroid (BFU-E) colonies (**A**,**B**), the wild-type and *Dot1l^KO^* myeloid colonies (CFU-GM) were similar in size (**C**,**D**).

**Figure 4 ijms-23-05137-f004:**
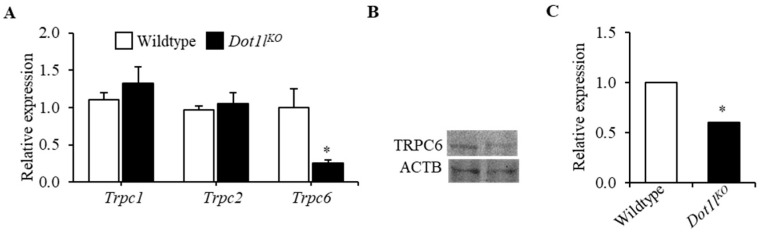
Expression of TRPC6 was markedly reduced in KIT-positive *Dot1l^KO^* hematopoietic progenitors. (**A**) Among the three genes in the *Trpc* family (i.e., *Trpc1*, *Trpc2*, and *Trp6*), only *Trpc6* levels were dramatically reduced in *Dot1l^KO^* compared to WT, while the levels of the other members remained relatively unchanged. TRPC6 protein levels in KIT-positive progenitor cells from *Dot1l^KO^* yolk sacs were also significantly reduced (**B**,**C**). Data are expressed as the mean ± SE; *n* > 6; * *p* < 0.05.

**Figure 5 ijms-23-05137-f005:**
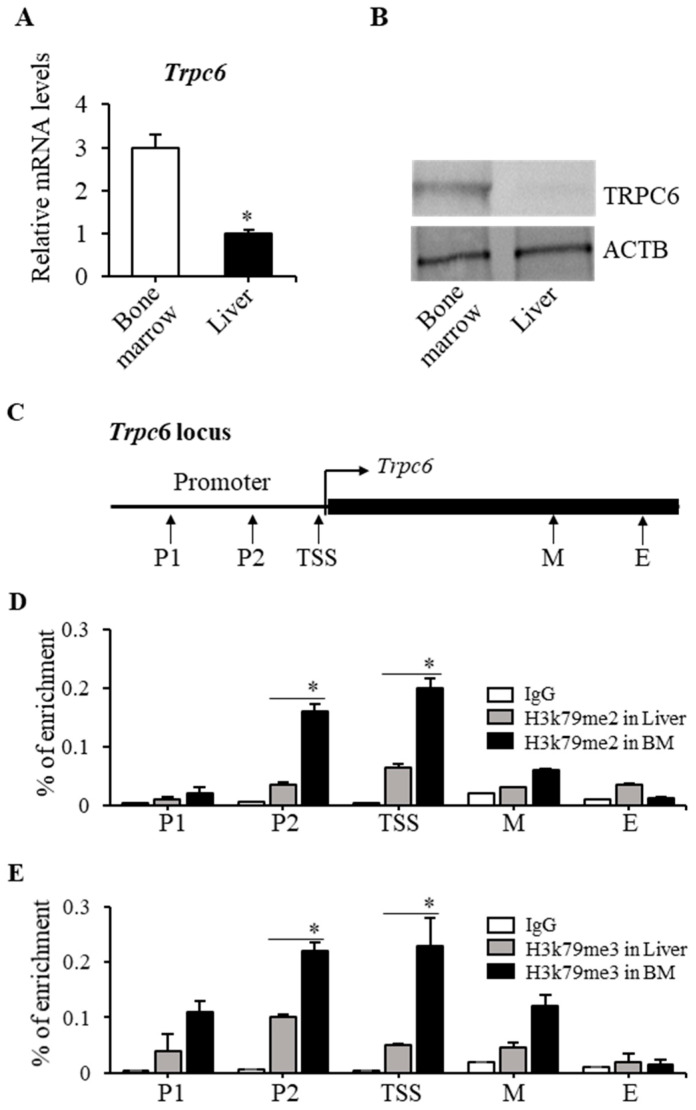
Enrichment of di- and tri-methyl H3K79 in *Trpc6* gene loci. *Trpc6* expression was tested by RT-qPCR in bone marrow (BM) and liver cells. *Trpc6* mRNA levels were significantly higher in BM cells compared to that in liver cells (**A**). TRPC6 protein was also detected in BM and liver cells by Western blot analysis, which showed that TRPC6 protein was also markedly higher in BM cells (**B**). ChIP assays were performed on chromatin preparations from BM and liver cells to assess H3K79 di- and tri-methylation in the *Trpc6* gene locus. (**C**) Several pairs of primers were designed covering the *Trpc6* gene locus including the promoter region (P1, P2), transcription start site (TSS), and middle and end of the gene locus (M, E). ChIP assays were performed in the promoter region of *Trpc6* gene locus for H3K79 di-methylation (**D**) and H3K79 tri-methylation (**E**). There was significantly greater enrichment of both H3K79 di- and tri-methylation at the P1, P2, and TSS sites of the *Trpc6* gene in BM cells than that of liver cells. The ChIP-qPCR data represent 3 independent experiments. Data are expressed as the mean ± SE; *n* > 6; * *p* < 0.05.

**Figure 6 ijms-23-05137-f006:**
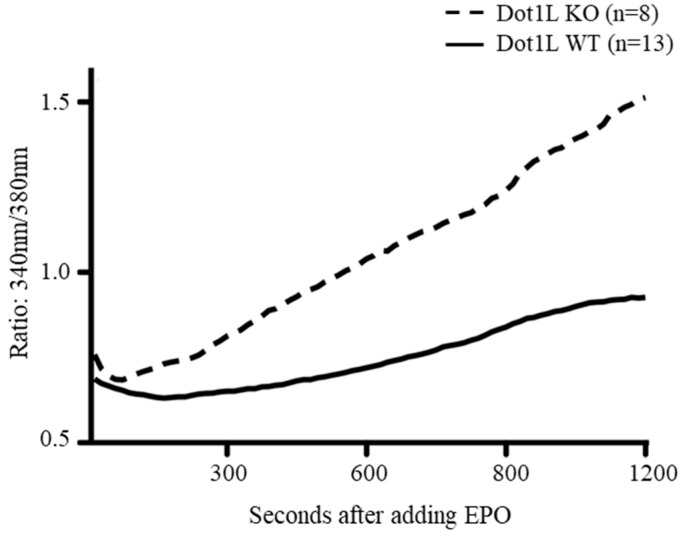
Accelerated and enhanced calcium influx in *Dot1l^KO^* erythroid progenitors. E10.5 *Dot1l^KO^* and wild-type yolk sac cells were stained, treated with EPO, and the Ca^2+^ signals in individual cells were recorded using fluorescence microscopy. After EPO treatment, progenitor cells from *Dot1l^KO^* yolk sacs showed a sustained increase in Ca^2+^ levels. Data are expressed as the mean ± SE; *n* > 6; **p* < 0.05.

**Figure 7 ijms-23-05137-f007:**
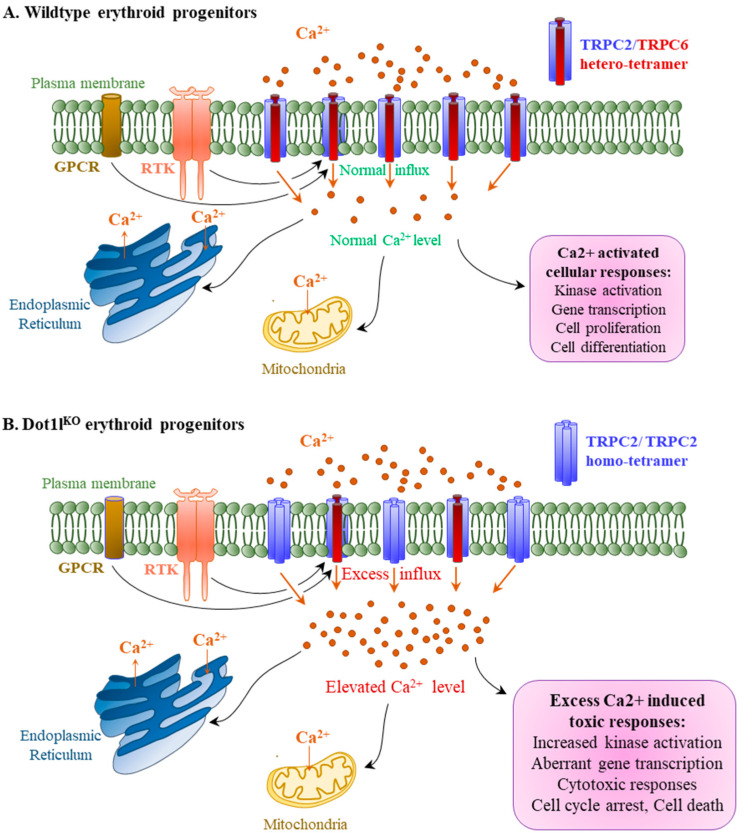
*Dot1l^KO^* erythroid progenitors exhibited increased calcium influx following decreased TRPC6 expression. TRPC2 allows Ca^2+^ influx into erythroid progenitors. TRPC6 forms a hetero-tetramer with TRPC2 and negatively regulates TRPC2-dependent Ca^2+^ influx to maintain the normal intracellular Ca^2+^ level (**A**). The loss of DOT1L decreases TRPC6 expression, resulting in an increased TRPC2/TRPC6 ratio. This favors the formation of TRPC2 homo-tetramers over TRPC2/TRPC6 hetero-tetramers leading to an excessive influx of Ca^2+^ into *Dot1l^KO^* erythroid progenitors (**B**). An elevated Ca^2+^ level results in toxic intracellular responses and aberrant cell signaling linked to cell cycle arrest and cell death observed in *Dot1l^KO^* erythroid progenitors.

**Table 1 ijms-23-05137-t001:** Primers used in the qRT-PCR studies.

Gene	Reference mRNA	Forward Primer	Reverse Primer
*Trpc1*	NM_0011643.4	366F: cggttgtcagtccgcagat	456R: tcgttttggccgatgattaagta
*Trpc2*	NM_011644.3_	631F: ctcaagggtatgttgaagcagt	741R: gttgtttgggcttaccacact
*Trpc6*	NM_004621.6	164F: gcttccggggtaatgaaaaca	255R: gtatgctggtcctcgattagc

**Table 2 ijms-23-05137-t002:** Primers for the ChIP-qPCR for the promoter, TSS, and intragenic sites within the *Trpc6* loci.

Target	Chromosome 9 Locus	Forward Primer	Reverse Primer
Promoter-1	8543340-680	aagcagggctcactgaatctgg	ggcattttccgatggtgtctg
Promoter-2	8543915-4113	cccaaataaagaatgtgcctgg	cgctgaagagttactatgtcaaccg
Trpc6 TSS	8544511-618	gagagccaggactatttgctgatg	tgccctcgcccatacttacaag
Trpc6 Middle	8549504-630	tctacctcctgatgctgggcttac	ggggtttgaagagatgagagtgc
Trpc6 End	8679716-902	tgccctacaaagcaatgaaagg	aaagagagcgtgagcccaacac

## Data Availability

SRA, NLM, USA (PRJNA666736).

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
