# Peer review of "DOT1L Methyltransferase Regulates Calcium Influx in Erythroid Progenitor Cells in Response to Erythropoietin"

_ijms, 2022, doi:10.3390/ijms23095137_

Round 1
Reviewer 1 Report
Major comments:
1. Effects of a chromatin modifier, such as DOT1L, must be analyzed on an epigenome-wide level and not only by ChIP-qPCR. Thus, without epigenome-wide data the publication is incomplete.
Minor comments:
- All abbreviations should be defined at the first time use and then applied consistently. This applies also to the Abstract.
- Please use for all gene and protein name abbreviation the latest HuGO nomenclature.
- Gene and mRNA name abbreviations should be in italic.
Author Response
Response to the queries of reviewer #1
We are thankful to the reviewer for her/his expert opinions. We have revised the manuscript to address the reviewer’s concerns and incorporated her/his suggestions. We appreciate that the reviewer’s suggestions have substantially improved the manuscript to make it acceptable for publication in the International Journal of Molecular Sciences. The modified or added sections are highlighted in the revised manuscript. Revisions that address the reviewer’s critiques are summarized below:
Query 1. Effects of a chromatin modifier, such as DOT1L, must be analyzed on an epigenome-wide level and not only by ChIP-qPCR. Thus, without epigenome-wide data the publication is incomplete.
Response to query 1. We agree with the reviewer that a genome-wide approach for DOT1L binding would provide a large amount of useful data. However, DOT1L is responsible for many cellular functions including cell proliferation, lineage determination, cell cycle regulation, and DNA damage repair. Although the primary function of DOT1L is H3K79 methylation, its regulatory functions can also be independent of H3K79. Accordingly, we can anticipate that the genome-wide data will not be limited to this study focus that deals with an abnormal erythropoietin signaling due to the decreased Trpc6 expression in yolk sac derived hematopoietic progenitor cells.
Our ChIP-qPCR data on methylated-H3K79 enrichment in Trpc6 gene loci indicate that a lack of H3K79 methylation (in the absence of DOT1L) in Dot1L-KO hematopoietic progenitor cells leads to decreased expression of Trpc6. We recently generated a mouse model that specifically lacks the methyltransferase activity of DOT1L protein (Malcom et al, 2022). RNA-seq analysis of the methylmutant yolk sac derived erythroblast cells shows a 9-fold downregulation (FDR p<0.00) in Trpc6 expression (Borosha et al, 2022), which is in line with our current data.
The premise of the study is an abnormal erythropoietin response due to the decreased Trpc6 expression in Dot1L-KO hematopoietic progenitor cells. This abnormal cellular response may play an important role in resulting the Dot1L-KO erythropoietic phenotypes (Feng et al, 2010; Malcom et al, 2022). However, our study does not claim that all the phenotypic abnormalities in Dot1L-KO yolk sac derived hematopoietic progenitors are due to Trpc6 downregulation. Therefore, a genomewide ChIP-seq data on DOT1L may place the readers focus out of the focus of this study.
To address the reviewers suggestion, we have added the following statements in the Discussion section of our revised manuscript:
“The premise in this study is that an abnormal EPO response occurs in Dot1lKO HPCs due to a decreased level of TRPC6 expression. Our data suggest that H3K79 methylation is disrupted in the absence of DOT1L, which is the underlying cause of decreased Trpc6 expression in Dot1lKO HPCs. We recently generated a mouse model that specifically lacks the methyltransferase activity of DOT1L protein [30]. RNA-seq analysis of the methyl mutant yolk sac-derived erythroblasts showed a 9-fold downregulation of Trpc6 expression (FDR p<0.00) [31] (PRJNA666736), consistent with the data presented in Figure 4. These data, coupled with our findings of enriched H3K79 di- and tri-methylation in the promoter and transcriptional start sites of the Trpc6 locus (Figure 5) indicate that DOT1L mediated H3K79 methylation is essential for Trpc6 expression in HPCs.
The decreased expression of TRPC6 increases the TRPC2/TRPC6 ratio. An increased TRPC2/TRPC6 ratio results in a prolonged and heightened level of Ca2+ influx in HPCs, in accordance with previously published findings [11]. We propose that the accentuated Ca2+ influx in response to aberrant EPO signaling plays an important role in determining the Dot1lKO erythropoietic phenotype [25,30]. However, our findings do not exclude the possibility that other DOT1L-regulated cellular mechanisms may contribute to the phenotypic abnormalities observed in Dot1lKO HPCs.”
Query 2. All abbreviations should be defined at the first time use and then applied consistently. This applies also to the Abstract.
Response to query 2. We have defined all the abbreviations at the first time throughout the manuscript including the abstract of our revised manuscript.
Query 3. Please use for all gene and protein name abbreviation the latest HuGO nomenclature.
Response to query 3. We have corrected all the gene and protein names in our revised manuscript according to the latest HuGO nomenclature.
Query 4. Gene and mRNA name abbreviations should be in italic.
Response to query 4. We are thankful to the reviewer. We have made the all the necessary corrections regarding gene and mRNA names.
Reviewer 2 Report
This manuscript describes the role of DOT1L in regulating TRPC6 expression in c-KIT positive erythroid progenitor cells. The authors found that a Trpc6 expression was notably down-regulated in DOT1LKO progenitor cells in the yolk sac. From the findings, the authors concluded that DOT1L directly targets Trpc6 in erythroid progenitor cells. Although the findings are informative, some points should be addressed as follows.
- The current introduction is too short of content. Authors should include more citations and content in the Introduction.
- The images in Figure (especially 2B and 3B) are not clear. Authors should improve the resolution of images.
- The location of all suppliers for equipment and reagents should be specifically addressed in the Materials and Methods section. (i.e. Sigma–Aldrich, St. Louis, MO, USA)
- Typological errors should be carefully checked and corrected.
Author Response
Response to the queries of reviewer #2
Overall comment: This manuscript describes the role of DOT1L in regulating TRPC6 expression in c-KIT positive erythroid progenitor cells. The authors found that a Trpc6 expression was notably down-regulated in DOT1LKO progenitor cells in the yolk sac. From the findings, the authors concluded that DOT1L directly targets Trpc6 in erythroid progenitor cells. Although the findings are informative, some points should be addressed as follows.
Response: We are thankful to the reviewer for her/his expert opinions. We have revised the manuscript to address the reviewer’s concerns and incorporated her/his suggestions. We appreciate that the reviewer’s suggestions have substantially improved the manuscript to make it acceptable for publication in the International Journal of Molecular Sciences.
The modified or added sections are highlighted in the revised manuscript. Revisions that address the reviewer’s critiques are summarized below:
Query/ Concern # 1. The current introduction is too short of content. Authors should include more citations and content in the Introduction.
Response to concern #1: We appreciate the reviewer’s suggestions. We have expanded the introduction section and incorporated all relevant literature in our revised manuscript.
Query/ Concern # 2. The images in Figure (especially 2B and 3B) are not clear. Authors should improve the resolution of images.
Response to concern #2: According to the reviewer’s suggestion, we have improved the image resolutions in all the revised Figures. However, due to requirement of a large number of new animals to obtain adequate numbers of KIT-positive yolk sac cells, we could not perform new western blots. If we need to perform new western blots, that will require additional 2-3 months.
Query/ Concern # 3. The location of all suppliers for equipment and reagents should be specifically addressed in the Materials and Methods section. (i.e., Sigma-Aldrich, St. Louis, MO, USA)
Response to concern #3: We have included all the manufacture’s names and addresses in revised manuscript.
Query/ Concern # 4. Typological errors should be carefully checked and corrected.
Response to concern #3: We are thankful to the reviewer. We have corrected all the typographical errors.
Round 2
Reviewer 1 Report
no additional comments